# Research on Dynamic Evolutionary Efficiency and Regional Differentiation of High-Tech Industrial Chain Networks

**Lihui Chen [1], Qiqi Xiao [2], Jianlin Wang [3] and Zhong Fang [2],\***

[1]    School of Marxism, Putian University, Putian 351100, China; ptxyclh@ptu.edu.cn
[2]    School of Economics, Fujian Normal University, Fuzhou 350117, China; qiqi15259113025@163.com
[3]    School of Finance, Fujian Business University, Fuzhou 350506, China; wangjianlin2023@fjbu.edu.cn
[*]    Correspondence: fazhong02@fjnu.edu.cn

**Abstract:** This research dynamically evaluates the innovation efficiency of China's high-tech industry and explores the efficiency differences in basic innovation, application innovation, and income innovation of this industry for different regions. Based on panel data of 30 provinces from 2015 to 2019, we construct a three-stage dynamic DDF (Directional Distance Function) model, divide the high-tech industry into three stages and measure the efficiency of the three stages. From 2015 to 2019, most of the total efficiency of China's high-tech industries showed an upward trend, with the western region having the highest total efficiency value and the central region the lowest. Most of the 30 provinces present basic innovation efficiency < applied innovation efficiency < profitable innovation efficiency. The conclusions from the article's empirical analysis can help developing countries concerned find out key links to improve the efficiency of high-tech entrepreneurship and innovation and then formulate relevant industrial policies.

**Keywords:** high-tech industry innovation efficiency; DEA model; three-stage network; dynamic efficiency

## 1. Introduction

Technological innovation efficiency refers to the ratio of the input of technological innovation resources to the output of technological innovation transformation. Higher output with less input indicates greater efficiency, while lower output suggests reduced efficiency [1,2]. Therefore, enhancing technological innovation resource input and optimizing technological innovation transformation output are key considerations in formulating government policy for high-tech industries in many developing countries [3]. However, the high-tech industry is distinguished from traditional industries by its extensive reliance on knowledge and technology, with significant investment in research, development and innovation. The intricacy of this technological transformation could result in regulatory lacunae for policymakers. With the acceleration of information technology and industrial revolutions in developing nations, these gaps in regulation present an increasing hurdle to sustainable development. During the "14th Five-Year Plan" period, high-tech industries driven by innovation have become the cornerstone of China's high-quality development. These industries have significantly contributed to the swift growth of China's economy in the recent past. By the end of 2018, statistics revealed that more than 181,000 high-tech enterprises existed in China, with an annual growth rate exceeding 15%. Moreover, the export value of high-tech products had reached 746.866 billion yuan. Furthermore, the proliferation of cutting-edge firms is being driven by state-of-the-art technology such as artificial intelligence and big data. In the present day, technological innovation is the main driving force behind the development of current industries and corporations. In the context of fierce worldwide rivalry in science and technology, the capacity for innovation is a crucial factor in the enhancement of developing nations' fundamental competitiveness.

Technological innovation efficiency is a key indicator of an enterprise's innovation capacity [4]. Due to the accelerating technological revolution and industrial transformation,

there is an discussion among Chinese scholars on the importance of increasing technological innovation efficiency and identifying bottlenecks within high-tech industries [5]. This is in response to the substantial growth in input and output in these industries. At present, China's high-tech industry R&D investment and R&D intensity continues to expand, but in the industrial cycle process, the innovation efficiency is low, the compound growth rate of R&D investment is significantly greater than the growth rate of new product sales revenue and other issues are very prominent, resulting in a large number of valuable resources being wasted [6]. These deficiencies make it somewhat difficult to improve the sustained competitiveness of the high-tech industry.

These limitations have a major impact on the continued competitiveness of the high-tech sector. Therefore, it is essential to employ a scientific approach to evaluate the various stages of the technological innovation cycle within the high-tech industry. Identifying potential areas for improving efficiency in technological innovation is crucial for addressing deficiencies. Pinpointing potential areas for improvement can greatly enhance the technological innovation efficiency. Valuable insights can be gained by scientifically assessing the different stages of the technological cycle in the high-tech industry chain. This is particularly relevant in the basic innovation stage, application innovation stage, and revenue innovation stage of the high-tech industry chain in various regions. Furthermore, it is imperative to address the technological disparity between regions in order to rectify the imbalance in regional development in China. This is crucial for the high-quality advancement of China's high-tech industries [7].

Based on the above analysis, this study presents a case study of China to investigate the intricate relationship between technological innovation efficiency in the basic innovation stage subsystem, the application innovation stage subsystem, and the revenue innovation stage subsystem in developing countries' high-tech industry chains. Furthermore, it evaluates the dynamic development efficacy of the mentioned subsystems while attempting to address the following key inquiries: (1) What objective methods can be used to evaluate the efficiency of technological innovations within China's high-tech industry? (2) To what degree do macroeconomic factors, including regional heterogeneity, impact the technological innovation efficiency within China's high-tech industry? (3) How can the technological innovation efficiency of China's high-tech industry chain be effectively improved to promote sustainable development and high-quality outcomes?

To tackle these issues, this study creatively devises a three-step index system for the fundamental innovation phase, application innovation stage, and revenue innovation stage in the high-tech industry. It also sets up a three-part chain network dynamic Data Envelopment Analysis (DEA) model. As a result, it obtains a three-stage non-radial DDF-DEA model. Additionally, to impartially demonstrate the technological innovation efficiency of each stage, this investigation integrates the non-radial Distance Directional Function (DDF) method into the conventional DEA model. Based on the empirical analysis of panel data from high-tech industries in 30 provinces in China between 2015 and 2019, this study enhances the comprehension of technological innovation's current situation in China's high-tech industries through dynamic evolutionary efficiency analysis. Additionally, it offers pertinent policy suggestions that other developing countries may use as a reference to boost technological innovation efficiency and attain sustainable and high standard growth.

The rest of the paper is organized as follows: the second part combs through the relevant literature on the evaluation of technological innovation efficiency in high-tech industries and puts forward the innovation pointed out in this paper. The third part describes the process of modeling as well as the sources and methods of dataset creation. The fourth part presents the results of the non-radial DDF-DEA three-stage model analysis. The fifth part analyzes the efficiency of the three-stage spatio-temporal evolution of the high-tech industry, and puts forward relevant policy recommendations for the current situation of technological innovation in China's high-tech industry for the reference of relevant developing countries.

## 2. Literature Review

The performance of the innovation efficiency of the high-tech industry is related to the modernization and transformation of the industrial structure of the high-tech industry, and the method of scientific evaluation of its innovation efficiency has obviously become a research hotspot of many scholars, and the research on the innovation metrics and evaluation of high-tech industry is mainly focused on the following two aspects:

(1) Research on factors influencing the innovation efficiency of high-tech industries. Some scholars believe that factors such as enterprise size, industry concentration, enterprise openness, government support, R&D investment, etc. will have a certain impact on the innovation efficiency of high-tech enterprises [8–11]. In recent years, some scholars have also explored the role of digital economy, green finance and other factors in improving high-tech industries, such as Ying et al. [12], who believe that the digital economy promotes the improvement of the industrial innovation efficiency of high-tech enterprises through strengthening entrepreneurship and industrial structure upgrading, and Wang [13], who argues that there is industry heterogeneity in terms of the impact of green financial policies on the R&D efficiency of technological innovations in high-tech industries.

(2) Research on evaluation of innovation efficiency of high-tech industries. Scholars at home and abroad mostly use hierarchical analysis, stochastic frontier analysis (SFA) and data envelopment analysis (DEA) to carry out assessments. Among them, the DEA method is more commonly used by many scholars, such as Ma et al. [14] who measured the innovation efficiency of listed companies based on the DEA method, and concluded that insufficient innovation investment is the main reason affecting the innovation efficiency. Peng et al. [15] assessed the innovation efficiency of science and technology enterprises in Hebei province in different regions and time nodes by constructing the SBM-Malmquist model. Some other scholars evaluate the innovation efficiency of high-tech industries by distinguishing the research samples, such as Kim et al. [16] and Liu et al. [17] who conducted empirical studies from the perspective of the structure of capital investment and from the perspective of the industry, respectively.

The above literature evaluates the innovation efficiency of high-tech industry based on different perspectives and methods, which have an important reference value for recognizing and assessing the innovation efficiency of the high-tech industry. However, an innovative scientific and technological achievement cannot be separated from technological success, product success and commercial success, and these methods are mainly based on the perspective of horizontal division, neglecting the efficiency assessment of the high-tech industry chain as a whole. Moreover, the above literature mostly evaluates the efficiency of the basic innovation stage, and does not take into account the evaluation of the efficiency of the subsequent stages such as the application and sale of new technologies. Therefore, some scholars began to focus on knowledge innovation, focusing on the commercialization of scientific and technological achievements as part of a two-stage innovation efficiency research, i.e., the first stage is the research and development stage, the emphasis is on invention; the second stage is the commercial transformation stage, focusing on scientific and technological innovations into products, commercialization and marketability, which in turn produce direct economic benefits. Xie et al. [18] and Xu et al. [19] divided the innovation process of high-tech industries into two stages, namely knowledge production and achievement conversion, based on the innovation chain perspective and concluded that the low efficiency of achievement conversion is an important factor hindering the enhancement of the green innovation efficiency in high-tech industries. Wang et al. [20] considered on this basis that the two-stage division of high-tech industry should focus on the correlation between the subunits of each stage and the overall unity of the technological innovation process. For this reason, they set up a chain-associated network DEA model and decomposed the process of the high-tech industry innovation system into two interrelated chain sub-processes, namely knowledge innovation and commercialization of scientific

and technological achievements, from the perspective of the two-stage value chain to carry out an efficiency evaluation. Although this method differs from the vertical perspective, it clearly fails to pay attention to the efficiency changes that may be brought about by the innovation transformation process from patent research and development to the marketization of new products.

Accordingly, from the perspective of the value chain, that is, the transformation of innovation resources input to output in the high-tech industry, this paper subdivides innovation in high-tech industry into three stages, namely basic innovation, applied innovation, and gainful innovation. Among them, the basic innovation stage consists of universities, scientific research institutes, high-tech research and development enterprises, etc., as the main innovation producers, undertaking basic innovation research and development; in the applied innovation stage, manufacturing enterprises, as the innovation consumers, buy new technologies developed by innovation producers for digestion and absorption, and then develop new products and services to turn to the market to realize the applied innovation; in the gainful innovation stage, the market and customers, as the new consumers and users of new products and services, assume the role of innovation disaggregators, and the enterprise obtains income by selling new products and services, successfully realizing product commercialization and marketization. Using the three-stage chain dynamic DDF model to include the intermediate output and final output of the high-tech industry under the same analytical framework for innovation efficiency evaluation, this paper makes up for the lack of existing literature on the assessment of the overall process of innovation in high-tech industry from the resource input to the transformation of the outputs, so as to obtain the evaluation results with a higher accuracy rate. On this basis, this paper puts forward targeted policy recommendations, which are of great significance to deepening the research on innovation efficiency evaluation and effectively improving the innovation capacity of China's high-tech industries.

## 3. Research Methods

Efficiency evaluation is a frequent and extremely important cognitive activity in human society. The evaluation of an economic management activity involves multiple indicators, which is a multi-agent and multi-path interactive process evaluation. This requires the evaluation method to balance multiple dimensions. Charnes et al. [21] proposed the CCR model that can be used to measure the performance of multiple inputs and multiple outputs at constant returns to scale, naming it the Data Envelopment Analysis (DEA). Banker et al. [22] presented the BCC model and revised the assumption of constant returns to scale into variable returns to scale. Tone [23] set up the Slack-Based Measure (SBM) model to modify the radial efficiency of the CCR model and the BCC model. At the same time, their model considers the slack between the input term and the output term and estimates the efficiency in a non-radial way. When considering an undesirable output, the DDF (Direction Distance Function) is a commonly used efficiency measurement tool. Chung et al. [24] first put forward the concept of a distance function based on the output orientation. Färe and Grosskopf and Chen et al. [25] established a non-oriented directional distance function, which provides a more reasonable and accurate estimation result.

Traditional DEA performs efficiency conversion through input and output, and the conversion process is regarded as a "black box". Fare et al. [26] offered Network Data Envelopment Analysis (Network DEA), whereby the production process is composed of many sub-production technologies, and the sub-production technologies are taken as sub-decision-making units (Sub-DMU). The CCR and BCC models are then used to find the most suitable solution. In order to analyze the efficiency of each sub-process, Chen and Zhu, Kao and Hwang, Kao [27], and Tone and Tsutsui [28] proposed a weighted SBM (weighted slack-based measures) network DEA model. The linkage between the various departments of the DMU is used as the analysis basis of the Network DEA model; each department is regarded as a Sub-DMU, and then the SBM model is used to find the most suitable solution.

Traditional DEA involves a static comparison and lacks any evaluation and analysis in different time periods. Kloop [29] first offered the window analysis as a form of dynamic analysis. Fare and Grosskopf [30] were the first to put inter-connecting activities into the model. Tone and Tsutsui extended the model to a dynamic analysis of slacks-based measures. Our study refers to the DDF model of Färe and Grosskopf [31] and Chen et al., plus the network DEA model of Färe et al. and the concept of dynamics, and proposes a three-stage dynamic direction distance function model. The results herein allow us to better understand the overall picture of China's high-tech industry innovation efficiency and avoids any underestimation or overestimation of efficiency.

### 3.1. Three-Stage DDF Dynamic Model

We assume that, due to different management types, resources, regulations, or environments, all manufacturers (N) are composed of decision-making units (DMUs) $(N = N_1 + N_2 + \ldots + N_g)$ of g groups. Suppose there are three stages in each t time period $(t = 1, \ldots, T)$: basic innovation stage, applied innovation stage, and profitable innovation stage. The basic innovation stage has I inputs $x_{ij}^t (i = 1, \ldots, I)$ to generate D intermediate products $z_{dj}^t (d = 1, \ldots, D)$ and K desirable outputs $q_{kj}^t (k = 1, \ldots, K)$. The applied innovation stage uses D intermediate products $z_{dj}^t (d = 1, \ldots, D)$ and G inputs $w_{gj}^t (g = 1, \ldots, G)$ to create desirable S output $y_{sj}^t (s = 1, \ldots, S)$ and E intermediate products $u_{ej}^t (e = 1, \ldots, E)$. The profitable innovation stage uses E intermediate products $u_{ej}^t (e = 1, \ldots, E)$ and B inputs $f_{bj}^t (b = 1, \ldots, B)$ to create desirable L output $n_{lj}^t (l = 1, \ldots, L)$, where $c_{hj}^t (h = 1, \ldots, H)$ is a carry-over factor.

The inputs in the first stage are R&D personnel and R&D expenditure, and the outputs are the number of invention patents and new product development, which are also the links from the first stage to the second stage. The second stage inputs are non-R&D personnel, non-R&D expenditures, new product development costs, and the outputs are the number of new projects and utility model patents, which are also the links from the second stage to the third stage. The inputs in the third stage are engaged personnel and operating expenses, the outputs are income from new products and main business income, and the carry-over variable is fixed assets.

As each DMU under the frontier chooses the most favorable final weighted output, the DMU efficiencies under the frontier are solved using the following equations.

#### 3.1.1. Objective Function

The efficiency of the DMU is:

$$max\ \theta = \sum_{t=1}^{T} \gamma_t (w_1^t \theta_1^t + w_2^t \theta_2^t + w_3^t \theta_3^t ) \tag{1}$$

s.t.

Basic innovation stage:

$$\sum_{j}^{n} \lambda_j^t X_{ij}^t \leq \theta_1^t X_{ip}^t \quad \forall i,\ \forall t$$

$$\sum_{j}^{n} \lambda_j^t z_{dj}^t \leq \theta_1^t z_{dp}^t \quad \forall d, \forall t$$

$$\sum_{j}^{n} \lambda_j^t q_{kj}^t \geq \theta_1^t q_{kp}^t \quad \forall k,\ \forall t$$

$$\sum_{j}^{n} \lambda_j^t \leq 1 \qquad \lambda_j^t \geq 0 \quad \forall j,\ \forall t \tag{2}$$

Applied innovation stage:

$$\sum_{j}^{n} \mu_j^t Z_{dj}^t \leq \theta_2^t Z_{dp}^t \quad \forall d, \forall t$$

$$\sum_{j}^{n} \mu_j^t y_{sj}^t \geq \theta_2^t y_{sp}^t \quad \forall s, \forall t$$

$$\sum_{j}^{n} \mu_j^t w_{gj}^t \leq \theta_2^t w_{gp}^t \quad \forall g, \forall t \quad \sum_{j}^{n} \rho_j^t n_{lj}^t \leq \theta_3^t n_{lp}^t \quad \forall l, \forall t$$

$$\sum_{j}^{n} \mu_j^t u_{ej}^t \leq \theta_2^t u_{ep}^t \quad \forall e, \forall t$$

$$\sum_{j}^{n} \mu_j^t = 1 \quad \mu_j^t \geq 0 \quad \forall j, \forall t \tag{3}$$

Profitable innovation stage:

$$\sum_{j}^{n} \rho_j^t u_{ej}^t \leq \theta_3^t u_{ep}^t \quad \forall e, \forall t$$

$$\sum_{j}^{n} \rho_j^t f_{bj}^t \leq \theta_3^t f_{bp}^t \quad \forall b, \forall t$$

$$\sum_{j}^{n} \rho_j^t n_{lj}^t \leq \theta_3^t n_{lp}^t \quad \forall l, \forall t$$

$$\sum_{j}^{n} \rho_j^t = 1 \quad \rho_j^t \geq 0 \, \forall j, \forall t \, \mu_j^t \geq 0 \, \forall j \, \rho_j^t \geq 0 \tag{4}$$

The link between the basic innovation stage and the applied innovation stage:

$$\sum_{j=1}^{n} \lambda_j^t Z_{dj}^t = \sum_{j=1}^{n} \mu_j^t Z_{dj}^t \quad \forall d, \forall t \tag{5}$$

The link between the applied innovation stage and the profitable innovation stage:

$$\sum_{j=1}^{n} \mu_j^t u_{ej}^t = \sum_{j=1}^{n} \rho_j^t u_{ej}^t \quad \forall e, \forall t \tag{6}$$

The link of the two periods:

$$\sum_{j=1}^{n} \lambda_j^{t-1} c_{hj}^t = \sum_{j=1}^{n} \lambda_j^t c_{hj}^t \quad \forall h, \forall t \tag{7}$$

Among the variables, $\gamma_t$ is the weight assigned to period t, while $w_1^t$, $w_2^t$, and $w_3^t$ are the weights assigned to basic innovation stage, applied innovation stage, and profitable innovation stage, respectively. Therefore, $w_1^t$, $w_2^t$, and $w_3^t \geq 1$ and $\sum_{t=1}^{T} \gamma_{tg} = 1$.

### 3.1.2. Period and Division Efficiencies

Period efficiency:

$$max\theta^t = w_1^t \theta_1^t + w_2^t \theta_2^t + w_3^t \theta_3^t \quad (t = 1 \dots 5) \tag{8}$$

Division efficiency:
Stage 1:

$$max\theta_1 = \sum_{t=1}^{T} \gamma_t(\theta_1^t) \tag{9}$$

Stage 2:

$$max\theta_2 = \sum_{t=1}^{T} \gamma_t(\theta_2^t) \tag{10}$$

Stage 3:

$$max\theta_3 = \sum_{t=1}^{T} \gamma_t(\theta_3^t) \tag{11}$$

### 3.2. Input, Desirable Output, and Undesirable Output Efficiency

We use Hu and Shieh's (2006) [32] total-factor energy efficiency index to overcome any possible biases in the traditional energy efficiency indicators, for which there are thirteen key efficiency models: R&D personnel, R&D expenditure, number of invention patents, new product development, non-R&D personnel, non-R&D expenditure, new product development costs, number of new projects, utility model patents, engaged personnel, operating expenses, new product sales income, and main business income. "I" represents area, and "t" represents time. The efficiency models are defined in the following.

$$\text{Input efficiency} = \frac{\text{Target input}}{\text{Actual input}} \tag{12}$$

$$\text{Undesirable output efficiency} = \frac{\text{Target Undesirable output}}{\text{Actual Undesirable output}} \tag{13}$$

$$\text{Desirable output efficiency} = \frac{\text{Actual Desirable output}}{\text{Target Desirable output}} \tag{14}$$

If the target inputs equal the actual inputs, then the efficiencies are 1, indicating overall efficiency; however, if the target inputs are less than the actual inputs, then the efficiencies are less than 1, indicating overall inefficiency.

If the target desirable outputs are equal to the actual desirable outputs, then the efficiencies are 1, indicating overall efficiency; however, if the target desirable outputs are more than the actual desirable outputs, then the efficiencies are less than 1, indicating overall inefficiency.

If the target undesirable outputs are equal to the actual undesirable outputs, then the efficiencies are 1, indicating overall efficiency; however, if the target undesirable outputs are less than the actual undesirable outputs, then the efficiencies are less than 1, indicating overall inefficiency.

## 4. Results and Discussion

### 4.1. Variables and Data

#### 4.1.1. Variable Interpretation

This study takes 30 provinces of China (including autonomous regions and municipalities) as the research objects. Based on the three-stage dynamic DEA model, we evaluate the basic innovation efficiency, applied innovation efficiency, and profitable innovation efficiency of these provinces.

We consider the different situations in each region of China. In the first stage, the basic innovation stage, R&D personnel and R&D expenditure are used as input variables, and the output variables are the numbers of invention patents and new product development. In the second stage, the applied innovation stage, this paper uses non-R&D personnel, non-R&D expenditure, and new product development cost as input variables, and the output variables are number of new projects and utility model patents. Among them,

invention patents and new product development are used as link indicators in the basic innovation stage and applied innovation stage, while fixed assets are selected as the carry-over indicator in the basic innovation stage. In the third stage, the profitable innovation stage, engaged personnel and operating expenses are used as input indicators, and new product sales income and main business income are output indicators. The number of new projects and utility model patents are indicators linking the applied innovation stage and the profitable innovation stage. The three-stage structure of the high-tech industry is shown in Figure 1.

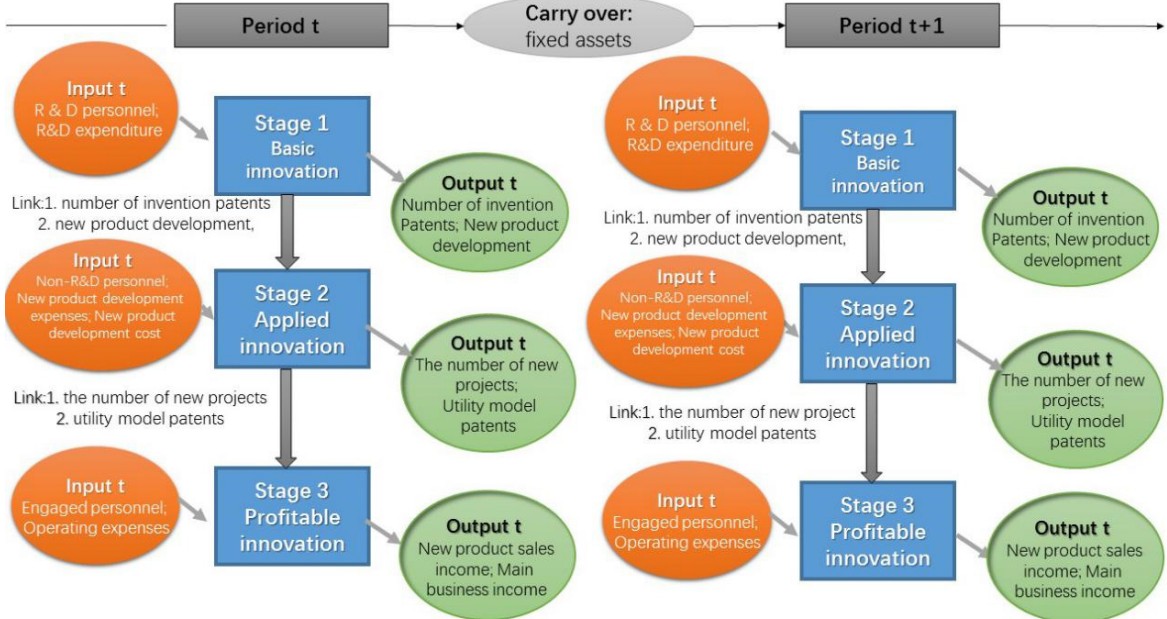

**Figure 1.** Three-stage structure of the high-tech industry.

The data on R&D personnel, R&D expenditure input, number of invention patents, new product development, non-R&D expenditure, new product development cost, number of new projects, and utility model patents are from the 2015–2019 China Science and Technology Statistical Yearbook, while that of non-R&D personnel, net engaged personnel, operating expenses, new product sales income, and main business income come from the 2015–2019 China Torch Statistical Yearbook and China High-tech Industry Statistical Yearbook. The specific variables are described in detail as follows:

(a)  R&D personnel are indicators used to measure the input of scientific and technological human resources, such as R&D full-time personnel (persons whose cumulative working hours engaged in R&D activities throughout the year account for 90% or more of their total working hours) workload and incomplete. The sum of the work is equivalent to the hourly personnel converted according to actual working hours.

(b)  R&D expenditure refers to the funds used for basic research, applied research, and experimental development. Specifically, it refers to the costs incurred by enterprises in the research and development process of products, technologies, materials, and processes.

(c)  The number of invention patents refers to the number of patents approved and authorized by the State Intellectual Property Office.

(d)  New product development is the actual number of product produced during the applied innovation research process.

(e)  Non-R&D personnel refers to scientific and technological personnel other than R&D scientists and engineers. This study uses year-end practitioners minus R&D personnel.

(f) Non-R&D expenditure is the expenditure of enterprises in the process of scientific research and experimental development for the introduction of technology, digestion and absorption, purchase of domestic technology, and technological transformation.

(g) New product development cost is usually measured by the amount of financial support in the process of new product development.

(h) The number of new projects reflects the actual project results after the investment.

(i) Utility model patents are the total number of new practical technical solutions proposed for the shape and structure of a product.

(j) Engaged personnel refers to the number of non-R&D personnel, minus the number of R&D personnel, from the total number of employees at the end of the year.

(k) Operating expenses refer to sales expenses—that is, various expenses incurred in the process of selling goods and materials and providing labor services. This study uses main business income minus total profit to measure operating expenses.

(l) New product sales income reflects the economic value of innovative products in the market.

(m) Main business income refers to income from the main business.

(n) Fixed assets are tangible assets held by the high-tech industry for the production of products, the provision of labor services, leasing, or operations and management. Their service life exceeds the end of one fiscal year.

### 4.1.2. Data Description

From the perspective of the maximum, minimum, and average value of R&D personnel, R&D expenditure, invention patents, new product development, non-R&D personnel, non-R&D expenditure and new product development cost, number of new projects, utility model patents, engaged personnel, operating expenses, new product sales income, and main business income, the eastern region is clearly larger than the central and western regions. In the basic innovation stage, the input indicators and output indicators in the eastern region have the highest values. In the applied innovation stage, the average value of utility model patents is the highest in the central region, while the western region is higher than the eastern region. In addition, the maximum value of utility model patents produced in the applied innovation stage in the western region from 2015 to 2019 was the highest of the three regions, and the minimum value in the western region from 2015 to 2019 (except for the number of new projects) has always been the highest among the three regions. The average, maximum, and standard deviation of all input and output indicators in the third stage of profitable innovation are the highest in the eastern region and in the western region. Taken together, the three stages in the eastern region have invested the most. The situation in the western region is exactly the opposite of the situation in the eastern region. The index values of the central region are between the eastern and western regions. This relates to the geographical environment, advantageous industries, technological development level, talent diversity, and regional policies of each region. The average, maximum, minimum, and standard deviation of the indicators in the three stages of the eastern, central, and western regions are shown in Table 1.

From the average values, the input and output indicators of the three stages in the eastern region are greater than those in the central and western regions. However, in terms of utility model patents, the average value of the central region is greater than that of the eastern region, and the western region is also higher than the eastern region. For all indicators in the basic innovation stage, the average value of the eastern region is about four times that of the central region, while the average value of these indicators in the western region is approximately 1/8 of that of the eastern region.

**Table 1.** Data description of input-output variables.

| Region | Statistical Indicators | R&D Personnel | R&D Expenditure | Number of Invention Patents | New Product Development | Non-R&D Personnel | Non-R&D Expenditure | New Product Development |
|---|---|---|---|---|---|---|---|---|
| China | Average | 26,064.78 | 1,056,169.74 | 11,917.85 | 3649.24 | 786,163.28 | 250,123.08 | 1,330,913.65 |
| | Max | 286,009.81 | 11,247,028.33 | 206,134.13 | 38,526.12 | 6,521,815.21 | 5,184,499.12 | 17,862,113.41 |
| | Min | 82.12 | 4143.13 | 13.21 | 24.32 | 30,740.31 | 1431.55 | 8853.12 |
| | St.Dev | 46,483.67269 | 1,911,188.295 | 32,664.40 | 6175.63 | 994,348.59 | 608,314.31 | 2,833,244.92 |
| East | Average | 54,183.28 | 2,219,495.47 | 27,404.35 | 7497.33 | 1,484,470.66 | 537,432.39 | 2,883,937.41 |
| | Max | 286,009.81 | 11,247,028.12 | 206,134.12 | 38,526.13 | 6,521,815.43 | 5,184,499.95 | 17,862,114.36 |
| | Min | 824.31 | 37,340.93 | 560.12 | 348.32 | 30,930.16 | 11,267.35 | 40,853.71 |
| | St.Dev | 67,324.29 | 2,760,685.61 | 50,382.06 | 8855.49 | 1,339,356.15 | 932,381.53 | 4,228,966.49 |
| Central | Average | 14,055.27 | 529,042.53 | 3917.46 | 2048.15 | 555,294.88 | 103,775.13 | 592,759.02 |
| | Max | 27,265.73 | 1,454,446.75 | 10,060.91 | 4695.17 | 1,179,966.88 | 415,383.54 | 1,598,880.53 |
| | Min | 2237.14 | 42,822.87 | 585.51 | 325.28 | 165,517.59 | 10,325.15 | 53,317.57 |
| | St.Dev | 8735.56 | 398,637.13 | 3191.72 | 1247.02 | 297,155.63 | 104,554.91 | 447,589.97 |
| West | Average | 6680.48 | 276,209.28 | 2249.85 | 965.58 | 255,760.17 | 69,248.65 | 314,729.66 |
| | Max | 33,144.33 | 1,394,285.13 | 13,761.25 | 4921.29 | 813,312.69 | 413,880.41 | 1,835,505.13 |
| | Min | 82.21 | 4143.33 | 13.14 | 24.92 | 30,740.41 | 1431.55 | 8853.94 |
| | St.Dev | 8916.84 | 372,305.42 | 3381.12 | 1172.36 | 218,699.44 | 102,769.33 | 456,710.45 |

| Region | Statistical Indicators | Number of New Projects | Utility Model Patents | Engaged Personnel | Operating Expenses | New Product Sales Income | Main Business Income |
|---|---|---|---|---|---|---|---|
| China | Average | 587.04 | 123,379.45 | 794,342.45 | 4763.78 | 17,044,869.66 | 5098.11 |
| | Max | 3452.21 | 960,233.31 | 6,688,894.12 | 44,405.93 | 208,509,695.92 | 46,747.47 |
| | Min | 14.13 | 1557.57 | 30,888.31 | 65.25 | 55,221.17 | 72.29 |
| | St.Dev | 677.69 | 161,574.37 | 1,008,475.72 | 8186.36 | 34,759,294.05 | 8671.22 |
| East | Average | 815.82 | 97,333.15 | 1,502,238.35 | 9323.41 | 35,820,460.95 | 9974.32 |
| | Max | 3452.48 | 732,163.59 | 6,688,894.25 | 44,405.93 | 208,509,696.54 | 46,747.32 |
| | Min | 14.25 | 3072.72 | 31,585.64 | 135.58 | 82,804.95 | 156.02 |
| | St.Dev | 942.09 | 134,182.07 | 1,359,471.55 | 12,010.89 | 51,467,767.95 | 12,695.12 |
| Central | Average | 705.94 | 192,699.58 | 559,041.64 | 2907.43 | 10,121,533.23 | 3119.3 |
| | Max | 1514.25 | 584,454.24 | 1,185,250.34 | 6957.23 | 36,536,924.11 | 7402.04 |
| | Min | 72.21 | 2953.34 | 165,039.36 | 373.81 | 387,785.19 | 418.08 |
| | St.Dev | 445.34 | 163,022.83 | 298,752.52 | 1855.01 | 10,321,747.14 | 1963.67 |
| West | Average | 271.77 | 99,011.13 | 257,574.41 | 1554.23 | 3,304,432.06 | 1661.69 |
| | Max | 991.19 | 960,233.31 | 829,788.14 | 6564.65 | 17,996,612.24 | 6942.89 |
| | Min | 22.12 | 1557.51 | 30,888.25 | 65.55 | 55,221.31 | 72.21 |
| | St.Dev | 267.37 | 173,146.77 | 220,778.71 | 1987.34 | 4,586,107.81 | 2093.37 |

Based on the difference in the average value of input and output indicators between the two regions, the basic innovation efficiency in the first stage of the eastern region is not much different from that of the central and western regions. Among all the indicators in the second stage, except for the output indicator of utility model patents, the central region is higher than the eastern region. The average value of the remaining indicators in the eastern region is the largest, and the average in the western region is the smallest. The efficiency value of the central region in some years in the second stage is higher than the eastern region. Among the input-output indicators in the third stage, the eastern region has the largest average value. Among profitable innovation efficiency, the eastern region has the highest overall efficiency value in 2015–2019.

The maximum values of input variables and output variables in the three stages in the eastern region are the largest among the three regions. The minimum of the input variables and output variables of the three stages in the central region is the largest among the three regions. The minimum value of investment indicators in the third stage of the western region is similar to that of the eastern region. In terms of profitable innovation, the minimum values of the east and the west are similar, but the maximum values are quite different. This also highlights the fact that the east has developed rapidly and the west has developed slowly. The maximum input and output of the central region in the three stages are all at the middle level. The basic innovation, applied innovation, and profitable innovation of the high-tech industries in the central region can be further improved substantially. In the basic innovation stage, the maximum and minimum values in the eastern, central, and western regions are the lowest, indicating that China's high-tech industries still have a lot of room for improvement in basic innovation efficiency.

The eastern region has the highest standard deviation among all the input and output indicators in the three stages, indicating that the eastern region has developed unevenly. The standard deviation of the central region is similar to the standard deviation of the western region, and the two regions have a larger standard deviation in new product sales income. In other indicators, the central region is smaller than the eastern region, and the profitable innovation efficiency of the central and western regions is lower.

*4.2. Analysis of Total Efficiency of the High-Tech Industry*

From 2015 to 2019, Table 2 shows the total efficiency values of the three stages of DEA from basic innovation, applied innovation, to profitable innovation in the high-tech industry in 30 provinces and cities of China.

From the three stages, the total efficiency of Beijing, Qinghai, Hainan, and Guangdong from 2015 to 2019 is 1, and their efficiency of technological innovation is the best. The efficiency values of the remaining provinces in the three stages from 2015 to 2019 showed an overall upward trend. Shanghai, Shanxi, Gansu, Anhui, Zhejiang, Hunan, Yunnan, and Guangxi showed a downward trend in the three-stage total efficiency values from 2015 to 2019. In most provinces, the average total efficiency of the three stages in five years is less than 1, presenting room for improvement.

The total value of the three-stage efficiency of Ningxia from 2015 to 2018 was 1, but it dropped to 0.9938 in 2019. In Ningxia 2019, from basic innovation through applied innovation to final profitable innovation, all levels of innovation have a downward trend. The total efficiency value of Chongqing from 2016 to 2017 was 1, 0.8041 in 2015, and 0.8675 in 2019, indicating that the level of innovation has improved.

Figure 2 compares the total efficiency of China's provinces from 2015 to 2019 and the distribution of the total efficiency of each province in five years. The radar chart clearly presents the efficiency gap between provinces in each year.

From Figure 2 we see that the total efficiency values of China's 30 provinces spread out radially, indicating that the overall score of 2015–2019 is getting better. The total efficiency value Beijing, Qinghai, Hainan, and Guangdong in the past five years is 1, and the overall score value of most provinces is in the 0.9 circle of the radar chart. Shandong, Gansu, Hebei,

Henan, Shaanxi, and Hubei have relatively low total efficiency values, and their total efficiency values in 2015–2019 are in the 0.6–0.8 range. Only Gansu showed a significant downward trend in the overall efficiency value during the five-year period, and the overall efficiency values of the remaining five provinces all increased. Among the provinces showing an upward trend, Inner Mongolia, Jiangxi, Guizhou, and Fujian have increased rapidly. Inner Mongolia and Jiangxi both rose by about 0.2, while Guizhou and Fujian rose by more than 0.3. In recent years, these four provinces have strengthened R&D personnel, R&D expenditure, and various innovation inputs in high-tech industries, which in turn have promoted the steady improvement of basic, applied, and profitable innovations.

**Table 2.** The total efficiency value of each province from 2015 to 2019.

| Rank | DMU | 2015 | 2016 | 2017 | 2018 | 2019 |
|---|---|---|---|---|---|---|
| 1 | Shanghai | 0.8317 | 0.9024 | 0.8526 | 0.7907 | 0.7921 |
| 2 | Shanxi | 0.9329 | 0.8443 | 0.8615 | 0.8796 | 0.8803 |
| 3 | Shandong | 0.7159 | 0.8215 | 0.7609 | 0.8029 | 0.8210 |
| 4 | Inner Mongolia | 0.6728 | 0.7249 | 0.9649 | 0.8531 | 0.8537 |
| 5 | Tianjin | 0.8096 | 0.8839 | 0.7937 | 0.8903 | 0.8903 |
| 6 | Beijing | 1.0000 | 1.0000 | 1.0000 | 1.0000 | 1.0000 |
| 7 | Sichuan | 0.7577 | 0.7861 | 0.7916 | 0.8468 | 0.8837 |
| 8 | Gansu | 0.9397 | 0.8889 | 0.7363 | 0.6582 | 0.6926 |
| 9 | Jilin | 0.9729 | 0.9653 | 0.9356 | 0.9901 | 0.9901 |
| 10 | Anhui | 0.8626 | 0.9545 | 0.8557 | 0.8260 | 0.8390 |
| 11 | Jiangxi | 0.8157 | 0.9963 | 1.0000 | 0.9920 | 0.9904 |
| 12 | Jiangsu | 0.7305 | 0.8189 | 0.7623 | 0.8720 | 0.8720 |
| 13 | Hebei | 0.6813 | 0.7062 | 0.6998 | 0.7308 | 0.7208 |
| 14 | Henan | 0.6962 | 0.6909 | 0.6807 | 0.7979 | 0.7692 |
| 15 | Qinghai | 1.0000 | 1.0000 | 1.0000 | 1.0000 | 1.0000 |
| 16 | Chongqing | 0.8041 | 1.0000 | 1.0000 | 0.9025 | 0.8675 |
| 17 | Hainan | 1.0000 | 1.0000 | 1.0000 | 1.0000 | 1.0000 |
| 18 | Zhejiang | 0.8776 | 0.8910 | 0.8579 | 0.8805 | 0.8477 |
| 19 | Shaanxi | 0.5701 | 0.5916 | 0.6291 | 0.7453 | 0.7329 |
| 20 | Hubei | 0.6459 | 0.7623 | 0.6677 | 0.7923 | 0.7829 |
| 21 | Hunan | 0.7760 | 0.6609 | 0.6509 | 0.7277 | 0.7302 |
| 22 | Guizhou | 0.7492 | 0.9639 | 0.8661 | 0.9766 | 0.9729 |
| 23 | Yunnan | 0.9974 | 1.0000 | 0.9817 | 0.9319 | 0.8923 |
| 24 | Heilongjiang | 0.8570 | 0.7290 | 0.7726 | 0.9804 | 0.9795 |
| 25 | Xinjiang | 0.9507 | 0.8114 | 0.8875 | 0.9966 | 0.9966 |
| 26 | Ningxia | 1.0000 | 1.0000 | 1.0000 | 1.0000 | 0.9938 |
| 27 | Fujian | 0.6137 | 0.7319 | 0.7965 | 0.9261 | 0.9484 |
| 28 | Guangxi | 0.9151 | 0.9551 | 0.9626 | 0.9112 | 0.9112 |
| 29 | Guangdong | 1.0000 | 1.0000 | 1.0000 | 1.0000 | 1.0000 |
| 30 | Liaoning | 0.7148 | 0.7855 | 0.8762 | 0.8481 | 0.8403 |

Figure 3 shows the changes in the total efficiency value of China's eastern, central, and western regions from 2015 to 2019. The total efficiency of the three stages of basic innovation, applied innovation, and profitable innovation in the western region is higher than that in the eastern and central regions. The five-year total efficiency value of the western region is above 0.85, the efficiency value is relatively high, and the innovation of high-tech industries is relatively good. The central region has the lowest overall efficiency value at around 0.8. The total efficiency value of the eastern region is in the middle position. The western region has the lowest input variables, but their efficiency value is higher than that of the central region. The level of innovation input and output in the western region is relatively high. The overall efficiency values of the three regions over the past five years have shown an upward trend. The total efficiency score of the western region from 2015 to 2019 shows a trend of increasing year by year. In 2017, the line graphs of the central region and the eastern region show an inflection point, and their total efficiency score fell to 0.8545 and 0.8031, respectively. While the overall trend is increasing, the efficiency value

lies below the western region. In 2018–2019, the total efficiency changes in the east, central, and west regions are approximately parallel.

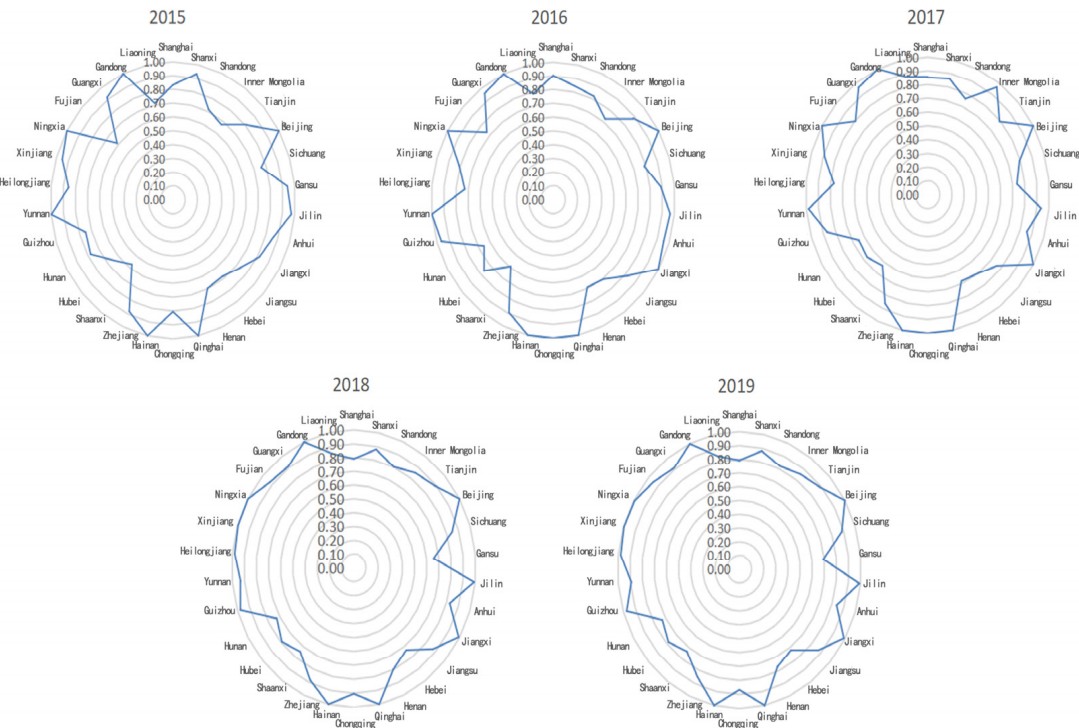

**Figure 2.** Radar chart of the total efficiency score of each province from 2015 to 2019.

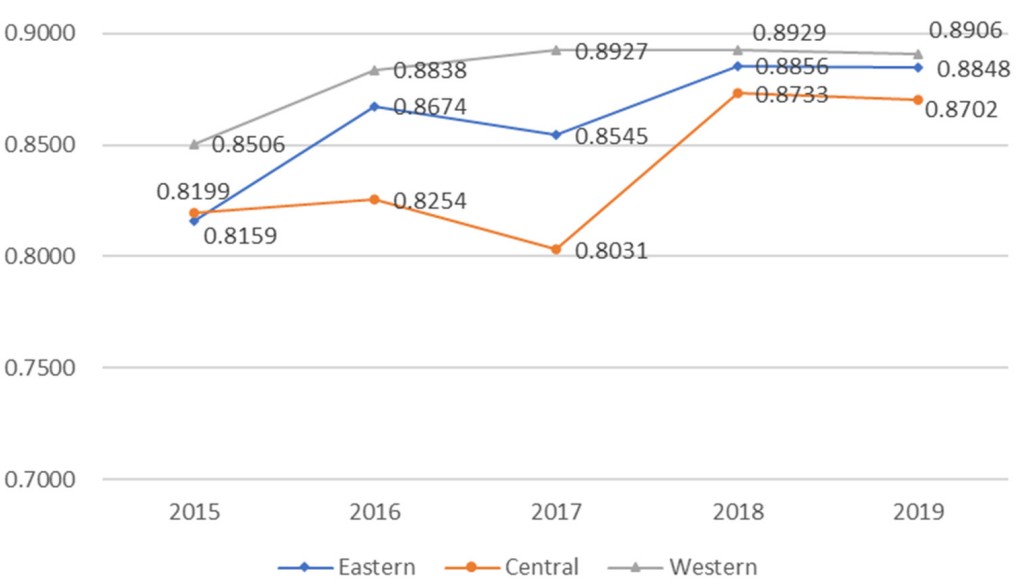

**Figure 3.** The overall efficiency of each region from 2015 to 2019.

*4.3. Analysis on Three-Stage Efficiency of the High-Tech Industry*

4.3.1. Basic Innovation Efficiency Analysis

The efficiency value of the first stage of the eastern region is higher than that of the central and western regions. Although the central region has the fewest provinces, its average production efficiency is the lowest. In the eastern region, the basic innovation efficiency value of Beijing, Guangdong, and Hainan from 2015 to 2019 is 1. It shows that the utilization of human, material, and financial resources for basic innovation investment in these provinces and cities is excellent. However, in the central region, there has not been

a province or city with a basic innovation efficiency score of 1 in the past five years. In the western region, only Qinghai has an efficiency score of 1 in the past five years.

As shown in Figure 4, the efficiency of basic innovation in the eastern region is higher than in the central and western regions with the central region as the lowest among the three. In the eastern region, except for Beijing, Guangdong, and Hainan, which have a basic innovation efficiency value of 1, in 2015–2019 the basic innovation efficiency value of most other provinces and cities has been on the rise (except Shanghai and Zhejiang). The basic innovation efficiency value of Shanghai in 2015 was 0.7141, after which the efficiency score continued to decline, dropping to 0.6978 in 2019. Zhejiang's basic innovation efficiency value dropped from 0.6329 in 2015 to 0.5430 in 2019. For the rising provinces, Fujian has the lowest basic innovation efficiency value of 0.36 in 2015, and it was also about 0.5 in 2016–2017, but it increased to 1 in 2018–2019. The efficiency value of basic innovation in Liaoning has been on the rise from 2015 to 2017.

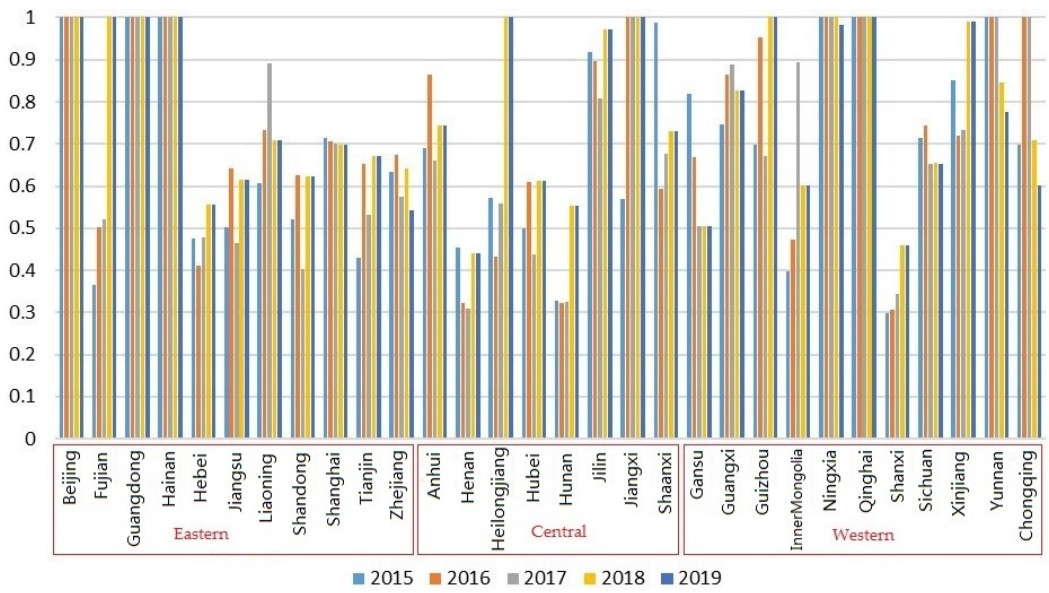

**Figure 4.** Basic innovation stage efficiency for 2014–2019.

The production efficiency value of the first stage in the central region is the lowest among the three regions. The basic innovation efficiency score of Jiangxi was about 0.5 in 2015, but significantly increased to 1 in 2016. Most of the remaining provinces have basic innovation efficiency scores between 0.5 and 0.8. The basic innovation efficiency values of Henan and Shanxi show a downward trend, while the other provinces are on the rise.

The basic innovation efficiency value of Qinghai in the western region from 2015 to 2019 was 1. The basic innovation efficiency value of Ningxia in 2015–2018 was 1, but dropped to 0.9813 in 2019. The efficiency scores of Shaanxi Province from 2015 to 2019 were all lower than 0.5, but show an upward trend. Xinjiang's basic innovation efficiency score rose from 0.8520 in 2015 to 0.9897 in 2019. The production efficiency value of Yunnan from 2015 to 2017 was 1, but dropped to 0.7765 in 2019. The basic innovation efficiency values of Gansu, Sichuan, and Chongqing also show a downward trend. The western region also has more room for improvement in terms of basic innovation. However, combined with the geographical environment and natural resources of the western region, the average basic innovation efficiency is higher than that of the central region, benefiting from the government's lasting role and follow-up replenishment.

4.3.2. Efficiency Analysis of the Applied Innovation Stage

Most of the applied innovation efficiency values in the three major regions of China are relatively high. Beijing, Guangdong, Hainan, Zhejiang, Jilin, Guangxi, Ningxia, and Qinghai have applied innovation efficiency values of 1 in 2015–2019. The applied innovation

efficiency of other provinces and cities in the past 5 years has shown an increasing trend, whereas that of Shanghai, Anhui, Hunan, Gansu, and Yunnan show a downward trend.

As seen in Figure 5, the efficiency of applied innovation in the eastern region is higher than in the central and western regions with the efficiency of applied innovation in the central region being the lowest. In the eastern region, Beijing, Guangdong, Hainan, and Zhejiang have applied innovation efficiency values of 1. The applied innovation efficiency value of most other provinces and cities has risen. However, the applied innovation efficiency value of Shanghai in 2015 was 0.8219, but dropped to about 0.7 in 2019. The applied innovation scores of the remaining provinces and cities in the eastern region have been in the range of 0.6–0.8 in the past five years, showing good performance.

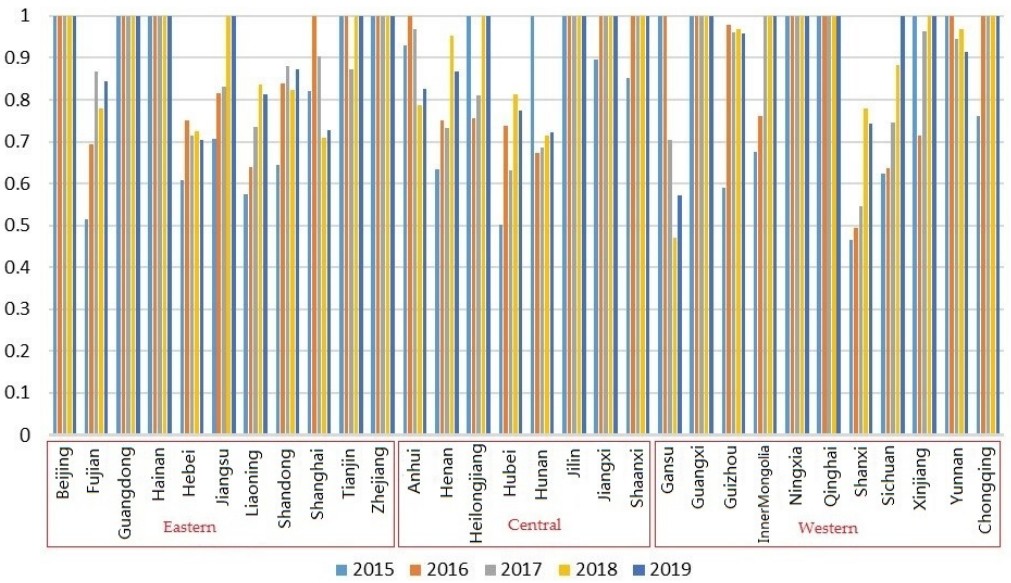

**Figure 5.** Applied innovation stage efficiency for 2014–2019.

Most of the provinces and cities in the central region have second-stage efficiency values from 2015 to 2019 that are presenting a growth trend. The applied innovation efficiency values of Anhui in 2015–2017 were all above 0.9, but the efficiency values in 2018–2019 were 0.79 and 0.83, respectively. Hunan had a low score for applied innovation efficiency in 2016–2019, peaking in 2015, while the efficiency of the remaining four years showed a downward trend. The efficiency values of most of the remaining provinces and cities are between 0.6 and 0.9. Overall, there is still room for improvement in applied innovation in the central region.

In the western region, Guangxi, Ningxia, and Qinghai have a 5-year efficiency value of 1, and the applied innovation efficiency is relatively high, reflecting the region's emphasis on applied innovation and support through industrial policies. However, the efficiency of applied innovation in Gansu continued to decline from 1 in 2015 to 0.5727 in 2019. Yunnan also dropped from its applied innovation efficiency score of 1 in 2015 to 0.92 in 2019. The applied innovation efficiency value of most of the remaining provinces shows an upward trend. There is a lot of room for improvement in the efficiency of applied innovation in the western region.

### 4.3.3. Efficiency Analysis of the Profitable Innovation Stage

The profitability innovation efficiency value of the three regions in China is relatively high, and the efficiency value of the third stage in all provinces is above 0.9. Among them, Beijing, Guangdong, Hainan, Zhejiang, Gansu, Henan, Jilin, Ningxia, and Qinghai all have profitable innovation efficiency values of 1. These provinces and cities have invested heavily in profitable innovation in high-tech industries and attached great importance to the innovative income level of these industries. Most provinces and cities show an upward

trend in their profitable innovation efficiency in the third stage over the 5 years. However, Hebei, Shandong, Shanghai, Anhui, Guangxi, Heilongjiang, Hunan, Jiangxi, Shanxi, and Yunnan show a downward trend.

As shown in Figure 6, the profitable innovation efficiency value of the eastern region is higher than that of the central and western regions. This relates to the high level of technological development, high talent density, and developed economy in the eastern region. Among them, the profitable innovation efficiency value of Beijing, Guangdong, Hainan, and Zhejiang from 2015 to 2019 is 1. The efficiencies of profitable innovation in Fujian, Jiangsu, and Liaoning all increased to 1 in 2019, while Hebei, Shandong, and Shanghai in the eastern region are on a downward trend. However, the profitable innovation stage performed better in the eastern region.

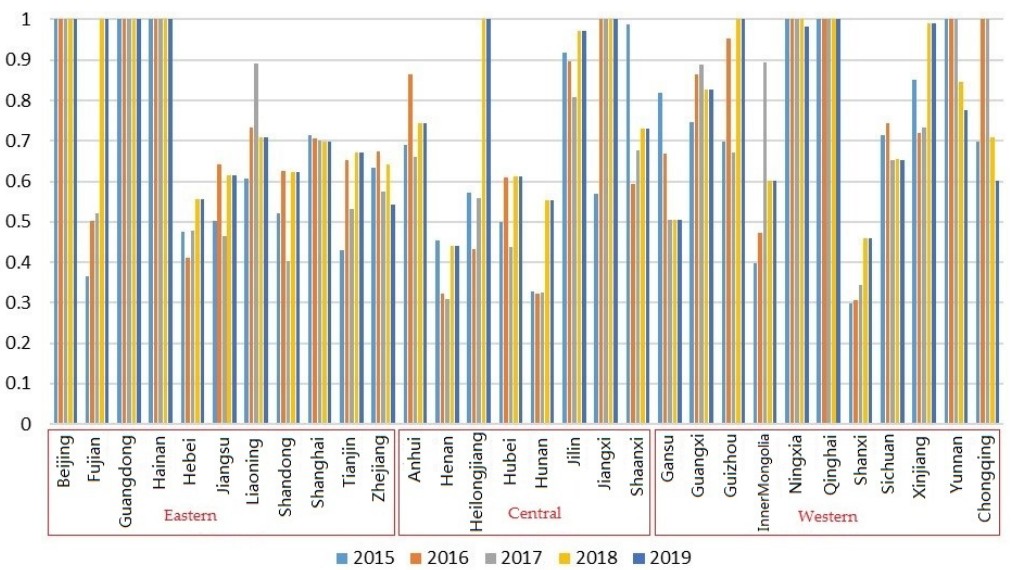

**Figure 6.** Profitable Innovation Stage for 2014–2019.

The profitable innovation efficiency value of the central region is only 1 in Henan and Jilin in 2015–2019. The efficiency of profitable innovation in Hubei has shown an upward trend, while the efficiency of profitable innovation in other provinces has declined. Therefore, the central region has a higher score for profitable innovation efficiency, but compared to other regions, its profitable innovation efficiency still has room for improvement.

The profitable innovation efficiency value of most provinces and cities in the western region (Guangxi and Yunnan) showed an upward trend in 2015–2019. The profitable innovation efficiency value of Gansu, Ningxia, and Qinghai from 2015 to 2019 is 1. These regions attach more importance to the level of profitable innovation in high-tech industries.

### 4.3.4. Comparison and Analysis of Three-Stage Efficiency of the High-Tech Industry

As a comparison, we note that basic innovation efficiency < applied innovation efficiency < profitable innovation efficiency. The exception is that the basic innovation efficiency and profitable innovation efficiency of Fujian were 1 in 2018–2019, but its applied innovation efficiency was lower than 1 in the past two years. The level of application of scientific and technological innovation in Fujian is relatively low. The basic innovation efficiency value and applied innovation efficiency value of Heilongjiang were 1 in 2018–2019, but its profitable innovation efficiency value was not 1 in 2018–2019 and showed a downward trend.

Jiangxi, Shanxi, and Guangxi show the efficiency of basic innovation < efficiency of profitable innovation < efficiency of applied innovation. This relates to the local high-tech industry policy, production economic activities, and operating standards. The efficiency of basic innovation and applied innovation of Jiangxi was 1 in 2016–2019, while its efficiency of profitable innovation was 1 in 2017, but was less than 1 in the other years. The applied

innovation efficiency of Shanxi was 1 from 2016 to 2019, but its profitable innovation efficiency has shown a downward trend in the past five years. The profitable innovation efficiency of Shanxi in 2019 was 0.91. The applied innovation efficiency of Guangxi was 1 in the 5 years, but its profitable innovation efficiency gradually declined from 2018 to 2019 with an efficiency value of 0.91.

The efficiency values of Beijing, Guangdong, Hainan, and Qinghai in the three stages from 2015 to 2019 were all 1. This correlates to their economic development level and technological innovation policies.

The dynamic three-stage efficiency comparison found that the efficiency value of the basic innovation stage is low, and the input-output effect of basic innovation on China's high-tech industry needs to be further improved. The efficiency values of the applied innovation stage and the profitable innovation stage are generally higher, which contribute to stronger overall efficiency. The profitable innovation efficiency of each province and city is the highest among the three stages. From 2015 to 2019, the high-tech industry innovation in various regions has brought better benefits, and the system of high-tech industry production and operation activities is mature.

Our comparison also finds that the efficiency of applied innovation > the efficiency of basic innovation, which means that basic research on high and new technology takes a long time to accumulate. The above analysis implies that the inputs of R&D personnel and R&D internal expenditures in the first stage are large, and basic innovation has a great impact on applied innovation.

The three-stage efficiency values of various regions in China have shown an upward trend, which means that the international competitiveness of its high-tech industries has increased in recent years. The efficiency of basic innovation is 0.40–0.70, the efficiency of applied innovation is mostly 0.60–0.90, and the efficiency of profitable innovation is mostly between 0.90–1.00. The basic innovation stage thus needs to be improved a lot more.

*4.4. Annual Efficiency Analysis of the Main Variables of the High-Tech Industry*

The efficiency values of the input and output variables in the three stages for China's 30 provinces from 2015 to 2019 are shown in Table 3.

The efficiency values of all input-output indicators in the three stages of dynamic DEA in Beijing, Guangdong, Hainan, and Qinghai from 2015 to 2019 are 1. The utilization efficiency of various variables is relatively high, and many resources have been fully and effectively used. This relates to the fact that Beijing is the national science and technology innovation center, Guangdong is the largest economic province in China, and these four cities are all vigorously developing high-tech industries.

The efficiency values of the input indicator R&D personnel and the output indicators of effective number of invention patents and new product development in the first stage of the basic innovation stage in most provinces in China are 1. The efficiency values of Shanghai, Yunnan, and Guangxi show a slight downward trend, while the efficiency values of the remaining provinces are on the upward trend. The provinces where the output efficiency is not 1 are showing a clear upward trend. The research and development efficiency of R&D personnel in the basic innovation stage is relatively high in all regions.

The efficiency values of non-R&D personnel and number of new projects in the second stage are mostly 1. In the provinces where the non-R&D personnel efficiency is not 1, Hebei and Henan are showing a downward trend, while the other provinces are all rising. In the third stage, the main business income efficiency score of almost all provinces is 1 (except for the efficiency value of Hunan in 2015–2019, which is 0.998).

The efficiency scores in the first stage of Beijing, Jiangxi, Henan, Qinghai, Hainan, Shaanxi, Hunan, Yunnan, Guangxi, and Guangdong are all 1 in 2015–2019. Inner Mongolia's R&D expenditure efficiency value is generally low and the average of efficiency value of Inner Mongolia's is 0.550 in 2015–2019. Ningxia's R&D expenditure efficiency value from 2015 to 2018 was 1, but dropped to 0.5384 in 2019. The efficiency value of most of the remaining provinces is between 0.8–0.95.

**Table 3.** The average efficiency of each indicator in each province in 2015–2019.

| DMU | R&D Personnel | R&D Expenditure | Non-R&D Personnel | Non-R&D Expenditure | New Product Development Cost | Engaged Personnel | Operating Expenses |
|---|---|---|---|---|---|---|---|
| Shanghai | 0.9891 | 0.9902 | 1 | 1 | 0.9592 | 0.9891 | 0.9901 |
| Shanxi | 0.8166 | 0.9286 | 0.9182 | 1 | 1 | 0.8166 | 0.9286 |
| Shandong | 1 | 0.8865 | 1 | 0.8371 | 0.8475 | 1 | 0.8865 |
| Inner Mongolia | 1 | 0.5502 | 0.9310 | 0.8687 | 0.9852 | 1 | 0.5502 |
| Tianjin | 0.9733 | 0.9532 | 1 | 1 | 1 | 0.9734 | 0.9532 |
| Beijing | 1 | 1 | 1 | 1 | 1 | 1 | 1 |
| Sichuan | 0.9646 | 0.8788 | 1 | 0.8542 | 0.9571 | 0.9647 | 0.8788 |
| Gansu | 0.9153 | 0.8752 | 0.9808 | 1 | 1 | 0.9153 | 0.8752 |
| Jilin | 0.9983 | 0.9869 | 1 | 1 | 1 | 0.9983 | 0.9869 |
| Anhui | 0.9651 | 0.9405 | 1 | 0.9713 | 0.9366 | 0.9651 | 0.9405 |
| Jiangxi | 1 | 1 | 1 | 0.8560 | 0.9400 | 1 | 1 |
| Jiangsu | 0.9601 | 0.9172 | 1 | 0.8340 | 0.7752 | 0.9601 | 0.9172 |
| Hebei | 0.9473 | 0.9577 | 0.9616 | 1 | 1 | 0.9473 | 0.9577 |
| Henan | 0.9051 | 1 | 0.9693 | 1 | 0.9822 | 0.9051 | 1 |
| Qinghai | 1 | 1 | 1 | 1 | 1 | 1 | 1 |
| Chongqing | 1 | 0.9774 | 1 | 1 | 0.9648 | 1 | 0.9774 |
| Hainan | 1 | 1 | 1 | 1 | 1 | 1 | 1 |
| Zhejiang | 0.9042 | 0.9812 | 1 | 1 | 1 | 0.9042 | 0.9812 |
| Shaanxi | 0.9662 | 1 | 1 | 0.7897 | 0.7905 | 0.9662 | 1 |
| Hubei | 1 | 0.8326 | 1 | 0.9961 | 0.8803 | 1 | 0.8326 |
| Hunan | 0.9469 | 1 | 1 | 0.7020 | 0.9284 | 0.9469 | 1 |
| Guizhou | 0.9532 | 0.9409 | 1 | 0.7043 | 0.9148 | 0.9532 | 0.9409 |
| Yunnan | 0.9785 | 1 | 0.9151 | 1 | 1 | 0.9785 | 1 |
| Heilongjiang | 0.9390 | 0.9774 | 1 | 0.9979 | 0.9492 | 0.9390 | 0.9774 |
| Xinjiang | 1 | 0.8327 | 0.8926 | 0.9380 | 1 | 1 | 0.8327 |
| Ningxia | 1 | 0.9077 | 1 | 1 | 1 | 1 | 0.9077 |
| Fujian | 0.9998 | 0.9417 | 1 | 0.4304 | 0.7355 | 0.9998 | 0.9417 |
| Guangxi | 0.9591 | 1 | 1 | 1 | 1 | 0.9591 | 1 |
| Guangdong | 1 | 1 | 1 | 1 | 1 | 1 | 1 |
| Liaoning | 0.9438 | 0.9619 | 0.9243 | 0.7574 | 1 | 0.9438 | 0.9619 |

| DMU | New Product Sales Income | Main Business Income | Number of Invention Patents | New Product Development | Number of New Projects | Utility Model Patents |
|---|---|---|---|---|---|---|
| Shanghai | 0.5986 | 1 | 1 | 0.9728 | 0.3407 | 0.6254 |
| Shanxi | 0.4874 | 1 | 1 | 1 | 1 | 0.8890 |
| Shandong | 0.9285 | 1 | 0.9746 | 1 | 0.9785 | 0.2055 |
| Inner Mongolia | 0.9841 | 1 | 0.8766 | 1 | 1 | 1 |
| Tianjin | 1 | 1 | 1 | 1 | 1 | 0.8071 |
| Beijing | 1 | 1 | 1 | 1 | 1 | 1 |
| Sichuan | 0.8157 | 1 | 1 | 0.9921 | 1 | 0.5412 |
| Gansu | 1 | 1 | 1 | 1 | 1 | 1 |
| Jilin | 1 | 1 | 1 | 1 | 1 | 1 |
| Anhui | 1 | 1 | 1 | 1 | 1 | 0.6967 |
| Jiangxi | 0.7170 | 1 | 0.9432 | 1 | 1 | 0.9133 |
| Jiangsu | 0.9298 | 1 | 0.9986 | 1 | 1 | 0.4314 |
| Hebei | 1 | 1 | 0.9518 | 1 | 0.5537 | 0.1499 |
| Henan | 1 | 1 | 0.9254 | 1 | 1 | 1 |
| Qinghai | 1 | 1 | 1 | 1 | 1 | 1 |
| Chongqing | 0.9882 | 1 | 0.8907 | 1 | 1 | 0.8373 |
| Hainan | 1 | 1 | 1 | 1 | 1 | 1 |
| Zhejiang | 1 | 1 | 0.9745 | 1 | 0.9411 | 0.8144 |
| Shaanxi | 0.7077 | 1 | 1 | 1 | 1 | 0.6088 |
| Hubei | 0.9787 | 1 | 1 | 0.9854 | 0.9285 | 0.2993 |
| Hunan | 1 | 0.9981 | 1 | 1 | 1 | 0.7885 |
| Guizhou | 0.6831 | 1 | 1 | 1 | 0.9355 | 0.0971 |
| Yunnan | 0.8410 | 1 | 1 | 0.9626 | 0.7846 | 0.7921 |
| Heilongjiang | 0.7211 | 1 | 0.9569 | 1 | 1 | 0.8323 |
| Xinjiang | 1 | 1 | 0.9940 | 1 | 0.8992 | 0.9444 |
| Ningxia | 1 | 1 | 1 | 1 | 1 | 1 |
| Fujian | 0.9946 | 1 | 0.9223 | 1 | 1 | 0.7822 |
| Guangxi | 0.7955 | 1 | 1 | 1 | 1 | 1 |
| Guangdong | 1 | 1 | 1 | 1 | 1 | 1 |
| Liaoning | 1 | 1 | 1 | 0.9159 | 0.6029 | 0.7549 |

From the second stage of non-R&D expenditure and new product development cost, the efficiency values of most provinces and cities are between 0.7 and 1.0. The difference in the efficiency value of new product development cost among regions in China is relatively

small, but the difference in the efficiency value of non-R&D expenditure is large. For example, Inner Mongolia, Jiangxi, Hunan, and Fujian have non-R&D expenditure efficiency values below 0.4 in some years. The non-R&D expenditure efficiencies of Inner Mongolia and Jiangxi were 0.3435 and 0.2798, respectively, in 2015, and the efficiency value was 1 in 2016–2019.

The efficiency value of the utility model patents in the second stage fluctuates greatly in each province. The efficiency values of Beijing, Gansu, Jilin, Henan, Qinghai, Hainan, Ningxia, Guangxi, and Guangdong are all 1 from 2015 to 2019. However, there are still many provinces with efficiency values below 0.3, such as Hebei and Shandong. This shows that the research and development of utility model patents in various provinces and in innovative applications are not yet mature and perfect.

In the third stage of engaged personnel, the efficiency value of Inner Mongolia, Hebei, Yunnan, Heilongjiang, and Xinjiang in 2015–2019 is 1. The efficiency value of engaged personnel in Inner Mongolia dropped significantly from 1 in 2015 to 0.4493 in 2019. Hebei Province from 0.9449 in 2015 to 0.4343 in 2019. This shows that the high-tech industries in Inner Mongolia and Hebei have insufficient engaged personnel. From the new product sales income, the efficiency value of most provinces is between 0.7–1.0. The efficiency value of new product sales income in the remaining provinces is on the rise. China's resource utilization efficiency is better in the stage of profitable innovation.

## 5. Conclusions and Policy Recommendations

### 5.1. Empirical Analysis Conclusions

This paper adopts the three-stage chain dynamic DDF model to calculate and analyze the input-output efficiency of 30 provinces, municipalities and autonomous regions in the three major regions of East, Central and West China. Therefore, a potential limitation of this study is that the obtained findings may be affected by data envelopment modeling (DEA) bias and confounding factors, which may affect the model estimates. This is because data envelopment modeling is among the non-parametric methods whose greatest limitation is that the treatment of some issues cannot be easily explained through economic principles. In addition, the nonparametric method cannot provide statistical inference, which makes the measurement results lack credibility, and since the nonparametric method will independently consider the influence of chance factors on output, etc., the efficiency level of its measurement is usually lower than the actual level. Moreover, non-parametric methods are very sensitive to the choice of indicators, and different choices of indicators may have a large impact on the measurement results, which also affects the credibility of the results. In the future research, the method discussed in the data envelopment model can be done through the method of linear programming by means of the restriction formula, and at the same time, further optimization can be carried out in the selection of the indicator system of high-tech industry, and the interval of the sample observation as well as the amount of data of the observation sample can also be increased.

(1)    From the three-stage total efficiency value of China's high-tech industries in various regions, Beijing, Qinghai, Hainan, and Guangdong have been 1 for five years. The overall scores of most provinces in the three stages in 2015–2019 are on the rise, but the overall efficiency values of Shanghai, Shanxi, and Gansu are showing a downward trend. The total efficiency value of the three stages from 2015 to 2019 in the western region is higher than that in the eastern and central regions. The central region has the lowest overall efficiency value among the three regions. The input variables in the western region are always the lowest, but its efficiency value is higher than that in the central region. The level of innovation input and output in the western region is relatively high.

(2)    From the three-stage efficiency, most of China's 30 provinces exhibit basic innovation efficiency < applied innovation efficiency < profitable innovation efficiency. However, Jiangxi, Shanxi, and Guangxi show that basic innovation efficiency < profitable innovation efficiency < applied innovation efficiency, which relates to local high-tech



industrial policies, production economic activities, and operating standards. The application innovation efficiency value of Fujian is lower than basic innovation and income innovation, which shows that the application level of scientific and technological innovation in Fujian needs to be improved. The efficiency of basic innovation and applied innovation of Jiangxi is 1 in 2016–2019, while its efficiency of profitable innovation is low. The efficiency of profitable innovation in Guangxi and Shanxi has shown a downward trend in the past five years. Beijing, Guangdong, Hainan, and Qinghai all have an efficiency value of 1 in the three stages in the past five years, which closely relate to their economic development level and technological innovation policies. The efficiency of basic innovation in the eastern region is higher than that in the central and western regions. The basic innovation efficiency of the central region is the lowest among the three regions. The input-output effect of basic innovation in China's high-tech industry thus needs to be improved. The efficiency values of the applied innovation stage and the profitable innovation stage are generally higher. This reflects that China's current high-tech industries are relatively complete, and its industrial operation activities are also relatively complete.

(3) From the first stage, the efficiency values of R&D personnel, number of invention patents, and new product development in the basic innovation stage of most provinces in China are 1. In addition to a slight decline in Guangxi and other regions, other provinces have shown an upward trend. The research and development efficiency of R&D personnel in China's high-tech industries is relatively high. In terms of R&D expenditure efficiency, the efficiency value of most provinces is between 0.8–0.9, and meaning that there is room for improvement in the R&D expenditure of China's high-tech industries.

(4) The efficiency values of non-R&D personnel and number of new projects in the second stage are mostly 1, but the efficiency of Hebei and Henan is showing a downward trend, while the efficiency of the remaining provinces is increasing. This shows that the efficiency of technology application in China's high-tech industries in various regions is relatively high. The non-R&D expenditure efficiency values of most regional input indicators in China are between 0.7 and 1.0, while the efficiency values of Fujian are below 0.4. The region has insufficient non-R&D expenditure input. The efficiency value of utility model patents has a large regional difference, and the efficiency value of some provinces is below 0.5. The research and development skills of utility model patents in innovative applications in China's provinces are not yet mature.

(5) The calendar year efficiency values of the input indicator operating expenses and the output indicator main business income of the third stage of gainful innovation stage in all provinces of China are all 1, and they are relatively stable, which reflects that China's high-tech industry currently plays a more significant role in economic development. In addition, in terms of the input variable engaged personnel, the fluctuation of the efficiency value of some provinces is larger, as the efficiency value of the engaged personnel in Inner Mongolia has significantly decreased from 1 in 2015 to 0.4493 in 2019. This indicates that the scale of the engaged personnel in high-tech industries in these regions cannot support their industrial development. In terms of the output variable, new product sales revenue, the efficiency value of most provinces is between 0.7 and 1.0, except for Shaanxi and Heilongjiang provinces, where the efficiency value shows a downward trend, and the rest of the provinces show an upward trend.

### 5.2. Policy Recommendations

Combined with the above analysis, it can be seen that there is still much room for improvement in basic innovation efficiency, applied innovation efficiency and income innovation efficiency in China's provinces. This paper will put forward the following countermeasure suggestions, which are also relevant to developing countries with similar resource endowments and industrial institutions:

(1) Investment should be increased in technological innovation, and basic innovation efficiency must be improved. At present, the overall level of basic innovation efficiency of China's high-tech industries is relatively low. It is thus necessary to improve the input mechanism for scientific and technological innovation, continuously broaden the investment and financing channels for high-tech industries, actively encourage and guide financial institutions and private funds to participate in scientific and technological innovation, establish a sound production–university–research cooperation development mechanism for high-tech industries, and introduce scientific researchers from universities or scientific research institutes to join the innovative activities of high-tech industries.

(2) The government should guide high-tech enterprises to strengthen independent innovation management and focus on the research and development of core technologies, make full use of public resources, give policy support to high-tech enterprises, guide enterprises to strengthen their focus on R&D departments, and give priority to R&D departments in terms of funding arrangements. At the same time, the government should pay attention to the cooperation and exchange between high-tech industries in various regions, promote coordinated development, and reduce the efficiency of high-tech innovation between regions. Regions with low innovation efficiency should learn from high-efficiency regions, make up for the shortcomings in their own development, and improve innovation efficiency. In terms of resource allocation, the central government should pay attention to the rational allocation and planning of resources, promote open cooperation among regions, and jointly build resource sharing mechanisms covering industry, finance, talents, technology, and information, so as to achieve complementary advantages while promoting regional coordinated development.

(3) For developing countries, they should tailor their policies to the local conditions and target the key issues that are dragging down their overall efficiency. Since there are big differences in resource factor endowment, economic development level and industrial structure of each country, the innovation efficiency of high-tech industry in each country has different key problems. For countries with low basic innovation efficiency, they should build a diversified basic innovation investment mechanism, increase government investment, encourage enterprises to play the main role in basic innovation, and widely absorb social funds. Meanwhile, due to the problems of large investment in basic innovation, long period of time, inconspicuous economic benefits, and strong externality, they should strengthen the international cooperation in the field of basic innovation in order to maximize the benefits. For countries with a low applied innovation efficiency value, it is necessary to provide the ability to apply science and technology innovation, optimize the innovation development environment, reduce the innovation cost of enterprises and improve the input-output ratio of innovation activities by promoting the construction of the legal environment, the basic knowledge and technology environment as well as the infrastructure environment such as road traffic. Countries with low innovation efficiency in high-tech industries need to focus on the industrial characteristics and local advantages of high-tech industries and on improving the technological income-generating efficiency of enterprises.

**Author Contributions:** Conceptualization, L.C. and Z.F.; Data curation, Q.X.; Formal analysis, Z.F.; Investigation, J.W.; Methodology, Z.F. and L.C.; Visualization, L.C.; Supervision, Z.F. and Q.X.; Project administration, Z.F.; Writing—original draft preparation, L.C. and Z.F.; Writing—review and editing, J.W. All authors have read and agreed to the published version of the manuscript.

**Funding:** This research was funded by Humanities and Social Science Youth Fund project of Ministry of Education (21YJC710008), Putian University introduces talents to start scientific research project (2023144), Major Projects of Fujian Social Science Base (FJ2020JDZ025, FJ2022JDZ022), Fuzhou Key Research Base of Social Sciences Min Merchants Research Center (2023FZB70) for financial support.

**Institutional Review Board Statement:** Not applicable.

**Informed Consent Statement:** Not applicable.

**Data Availability Statement:** Data are contained within the article.

**Conflicts of Interest:** The authors declare no conflict of interest. The funding sponsors had no role in the design of the study, the collection, analyses, or interpretation of data, the writing of the manuscript, or in the decision to publish the results.

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
