# Peer review of "Research on Dynamic Evolutionary Efficiency and Regional Differentiation of High-Tech Industrial Chain Networks"

_sustainability, doi:10.3390/su152416643_

Round 1
Reviewer 1 Report (Previous Reviewer 3)
Comments and Suggestions for Authors
The paper can be accepted after following modifications.
-There is a problem with Eq. (11). Some characters of the nominator and denominator cannot be read.
- Please use the same number of decimals for each number written in the same column in tables.
Author Response
Please see the attachment, Thanks!

Reviewer 2 Report (New Reviewer)
Comments and Suggestions for Authors
I liked the research idea, and the methodology used to reach its conclusions is sound. I appreciated in the paper the fact that the authors uncovered an interesting area of analyzing the efficiency and regional differentiation of high-tech industrial chain networks in a dynamic context, and their data set is compelling and could be further analyzed. However, in an academic journal I expect to see novel theoretical contributions, and this paper, although well-developed and competently written, does not have a significant contribution to the field. I would recommend the authors extend their work by trying to propose their own model or framework and contrast it with those already available in the literature. The paper's methodological apparatus is appropriate to the task at hand and supported by academic literature. I believe this is a very interesting read and I congratulate the authors for their efforts, but I believe that it probably belongs to a different type of publication, focused on real-life applications.
Comments on the Quality of English LanguageThe phrasing is a bit combersome, and makes the paper difficult to follow at times. Probably a native English speaker could help improve the general phrasing of the paper.
Author Response
Please see the attachment,Thanks!

Round 2
Reviewer 2 Report (New Reviewer)
Comments and Suggestions for Authors
This is an interesting paper, which gives a general view evaluates the innovation efficiency of China’s high-tech industry and explores the efficiency differences in different forms of innovation for different regions. Although it relies on secondary data, the paper manages to find some interesting effects that manage to spring an interesting discussion. I believe that the last part of the analysis is slightly forced and the effects observed can be explained by various other variables that have not been taken into account. On the hole I believe this is an interesting paper and it raises interesting points and avenues for further research.
Comments on the Quality of English LanguageSome rephrasing is also recommend, as the paper shows that English is not the native language of the authors.
Author Response
Please refer to the attachment,Thanks!

This manuscript is a resubmission of an earlier submission. The following is a list of the peer review reports and author responses from that submission.
Round 1
Reviewer 1 Report
Comments and Suggestions for Authors
The word "Analysis of the" should be removed from the title.
Provide the full meaning of acronyms before they are used in the study.
The bigger problem of the study is the use of long sentences. Most of the sentences are so long and difficult to understand.
The research problem should be discussed.
Provide the organization of the study.
Include recent studies and cite them accordingly.
summary of literature should be provided.
What are the merits and demerits of the selected empirical model
Expose the economic implications of the study.
Discuss the limitation of the study and direction for improvements.
Comments on the Quality of English LanguageThe study will benefit from language editing.
Author Response
Please see the attachment, Thanks!

Reviewer 2 Report
Comments and Suggestions for Authors
1. Title and theme of the paper is justified. But title needs revision. Make a scientific title following the call for paper of this journal.
2. Overall structure of thematic content is OK. In my opinion, authors have taken a limited approach to measure and study cleaner environment. Authors must take a multidimensional approach to study natural resources and green economic recovery that need to be improved.
3. The abstract of the manuscript is defined and precise but still there is a room to improve it. However, it should discuss the finding of the study along with policy implications in a way that reader generates interest to read the whole paper. If possible, please add 1-2 lines about policy implications in the abstract.
4. Introduction section requires to explain the novelty of the paper including study motivation, contribution and research problem. Please try to revise the contribution and study motivation paragraph with brief and comprehensive detail.
5. The contribution seems dim in this manuscript. I would suggest the author to enhance your theoretical discussion and arrives your debate or argument to show satisfactory contribution. Refine research contribution and objectives. Clearly state both theoretical and practical contributions.
6. The authors have failed to explain the gap for the current study, although on page 6 they have mentioned a brief summary, but just naming some techniques is not sufficient to highlight the importance of current study. Please revise.
7. Rectify syntax errors and grammatical issues in it. Particularly, you can improve the advantages of this and its application with broader context.
8. The policy response of different countries should be explained in detail along with the policy recommendations.
Comments on the Quality of English Language1. Title and theme of the paper is justified. But title needs revision. Make a scientific title following the call for paper of this journal.
2. Overall structure of thematic content is OK. In my opinion, authors have taken a limited approach to measure and study cleaner environment. Authors must take a multidimensional approach to study natural resources and green economic recovery that need to be improved.
3. The abstract of the manuscript is defined and precise but still there is a room to improve it. However, it should discuss the finding of the study along with policy implications in a way that reader generates interest to read the whole paper. If possible, please add 1-2 lines about policy implications in the abstract.
4. Introduction section requires to explain the novelty of the paper including study motivation, contribution and research problem. Please try to revise the contribution and study motivation paragraph with brief and comprehensive detail.
5. The contribution seems dim in this manuscript. I would suggest the author to enhance your theoretical discussion and arrives your debate or argument to show satisfactory contribution. Refine research contribution and objectives. Clearly state both theoretical and practical contributions.
6. The authors have failed to explain the gap for the current study, although on page 6 they have mentioned a brief summary, but just naming some techniques is not sufficient to highlight the importance of current study. Please revise.
7. Rectify syntax errors and grammatical issues in it. Particularly, you can improve the advantages of this and its application with broader context.
8. The policy response of different countries should be explained in detail along with the policy recommendations.
Author Response
Please see the attachment, Thanks!

Reviewer 3 Report
Comments and Suggestions for Authors
Thanks for your efforts on preparation of this interesting paper. It shows a good scientific value, however some revisions are recommended before a publication decision. The points I can suggest are as follows:
- Introduction part should contain an emphasis on applied methodology for the problem. What kind of methods were/can be used in similar problems and why did you choose the one applied in the manuscript? It must be included into the introduction part.
- Introduction part needs a paragraph providing information about manuscript organization. Please add a last paragraph to eliminate this missing content.
- The last publication time for the references is 2020 in the manuscript. You should expand the literature review and refer more recent publications in literature review and other parts.
- Methodology explanations may contain a short expression of efficiency analysis before the basic information of DEA models.
- Why didn't you assign equation numbers for equations in lines 175-178?
- Verbal explanations for the model equations should be given.
- Sign restrictions for decision variables must be provided.
- Variable column in table 1 may only contain variable numbers given with description in lines 250-284. In the current form it is very complex to read.
- In line 425 there is miswritten word 22015. A complete check of the manuscript may be good.
- The discussion for study results seems good, however limitations and further research suggestions are missing. Please provide them in the conclusion part.
Comments on the Quality of English LanguageA spell check may be needed to eliminate language errors.
Author Response
Please see the attachment, Thanks!

Round 2
Reviewer 3 Report
Comments and Suggestions for Authors
I see the revision letter and revised form of the manuscript to be inelaborately prepared. Based on the decision of the editor; if you are requested to send a revised paper, you should prepare a more improved form of the revised manuscript to remove my concerns #4, #5, #6, #8, #10 and an elaborately prepared response letter . Your responses in the letter doesn't correspond to the revised manuscript and some responses provided are irrelevant. It is not a good thing for a scientist.
Comments on the Quality of English LanguageThere are punctuation and writing mistakes. A proofreading is recommended.